# Smart Electrically Assisted Bicycles as Health Monitoring Systems: A Review

**DOI:** 10.3390/s22020468

**Published:** 2022-01-08

**Authors:** Eli Gabriel Avina-Bravo, Johan Cassirame, Christophe Escriba, Pascal Acco, Jean-Yves Fourniols, Georges Soto-Romero

**Affiliations:** 1Laboratory for Analysis and Architecture of Systems (LAAS), University of Toulouse, F-31077 Toulouse, France; cescriba@laas.fr (C.E.); pacco@laas.fr (P.A.); fourniol@laas.fr (J.-Y.F.); gsotorom@laas.fr (G.S.-R.); 2EA4660, Culture, Sport, Health and Society Department and Exercise Performance, University of Bourgogne-France Comté, 25000 Besançon, France; johancassirame@free.fr; 3EA7507, Laboratoire Performance Santé Métrologie Société, 51100 Reims, France; 4Société Mtraining, R&D Division, 25480 Ecole Valentin, France

**Keywords:** e-bike, health, monitoring systems, intelligent sensors, physical activity, physiology

## Abstract

This paper aims to provide a review of the electrically assisted bicycles (also known as e-bikes) used for recovery of the rider’s physical and physiological information, monitoring of their health state, and adjusting the “medical” assistance accordingly. E-bikes have proven to be an excellent way to do physical activity while commuting, thus improving the user’s health and reducing air pollutant emissions. Such devices can also be seen as the first step to help unhealthy sedentary people to start exercising with reduced strain. Based on this analysis, the need to have e-bikes with artificial intelligence (AI) systems that recover and processe a large amount of data is discussed in depth. The Preferred Reporting Items for Systematic Reviews and Meta-Analyses guidelines were used to complete the relevant papers’ search and selection in this systematic review.

## 1. Introduction

In recent years, electrically assisted bikes have become more popular as a way to achieve active mobility or for recreation. In the United States, sales increased by 145% from 2019 to 2020, surpassing regular bikes with a 65% growth rate [1]. In Europe, the market increases by 23% each year, and it is estimated that by 2030 there will be twice as many e-bikes as cars [2]. It is well known that e-bikes are an excellent way to engage in physical activity, allowing users to travel on roads that might be too demanding (with challenging up-hill segments). They also reduce the user’s commuting time and carbon footprint [3,4]. Even more now, with the COVID-19 pandemic, people avoid using public transportation (because of social distancing) for their travels [5].

In the literature many articles evaluate the potential of electric bikes as an effective active mobility tool that can aid many to meet their physical activity guidelines. Bourne et al. [6] compared several studies showing the e-bike health potential by analyzing physical activity outcomes between regular bikes and walking. Besides, they made several comparative effort tests using no electric assistance, low and high assistance, and traveling on different types of roads (e.g., flat, hilly routes). As a result, even though the amount of exercise performed was smaller than amounts performed on a regular bike, it is still a better alternative to cars and public transportation; the lower perception effort could incentivize more people to use e-bikes as their daily means of transportation.

The World Health Organization (WHO) recommends a minimum of 150 min of moderate-intensity physical activity or 75 min of vigorous-intensity, which leads to numerous benefits and prevents chronic diseases [7].

Chronic diseases are major causes of death in the world (see Figure 1); WHO [8] has reported that seven out of these ten causes of death are from chronic (non-communicable) diseases, and are responsible for 44% of deaths. All chronic diseases together are responsible for 74% of deaths globally in 2019. Smoking, inadequate nutrition, lack of physical activity, and alcohol are the leading causes of cardiovascular diseases, cancer, chronic respiratory disease, and diabetes [9,10].

Booth et al. [12] examined 35 chronic diseases and their relationship to physical inactivity; they clarify the usage of physical activity, prevention of disease being one, and it is primarily used as a means to avoid physical inactivity, which is the direct cause of certain diseases. Physical activity is used as a treatment principally to diminish the side effects of pathologies. Pedersen and Saltin [13] recompiled the impact of physical activity as a therapy in 26 chronic diseases, in which for each disease they detailed the background, evidence-based physical training, possible mechanism, type of training and contraindications. Lee et al. [14] concluded that over the years the life expectancy of the physically active would increase by 0.68 years. Meanwhile, if inactive individuals started performing an activity, they would gain from 1.3 to 3.7 years (in the United States). This is also noticeable in Asia, where active people live 2.6 to 4.2 years longer than inactive ones.

Such health facts denote the importance of having intelligent electric bikes that could supervise their riders by collecting physiological, mechanical, and environmental data. This data helps with making some optimal decisions, such as establishing adequate electrical assistance, battery management, and sending information to peers (e.g., research team, medical staff, and coaches). Considering the above-mentioned, this review compares the existing systems that monitor the user’s physiological signals and analyzes the data in order to control the bike. In addition, this study discusses the impacts on health that the existing systems could have on users with reduced mobility or pain difficulties (chronic diseases, knee and ankle procedures, sedentary habits).

## 2. Materials and Methods

### 2.1. Search Procedure

This review was done under the Preferred Reporting Items for Systematic Reviews and Meta-Analyses (PRISMA) guidelines, Liberati et al. [15]. The following databases were consulted in March 2021: Web of Science, PubMed, IEEE, Science Direct, Google Scholar, and HAL (France open archives), ending the search task in June 2021. The keywords used to consult each database, including their synonyms and combinations, were: e-bike, electrically assisted bike, electrically assisted bicycle, health, and monitoring systems. Thus, additional filters focused this search of multi-sensing e-bikes and their application within the medical world (e.g., in Web of Science Categories: ENGINEERING ELECTRICALELECTRONIC OR MEDICINE GENERAL INTERNAL; on Science Direct’s publications titles: Medical Engineering & Physics or Procedia Engineering).

### 2.2. Inclusion Criteria

Some articles addressed the fundamental aspects of an electric bike (see Figure 2), such as motor control (i.e., field-oriented [16,17], fuzzy logic [18,19,20], firefly algorithm [21], particle swarm optimization [22], model predictive [23], reinforcement learning [24], and others [25,26,27,28,29,30,31]) using different inputs, such as torque (human and machine), power and speed to control the bike’s motor. Studies on battery management system (BMS) and energy recovery [18,23,31,32,33,34], explored methods of supervising and charging the batteries used by the bikes (State of Health and State of Charges) and also explored the possibility of recovering energy by braking and in downhill situations. These systems need to work alongside the motor controllers to extend the battery at disposal and extend the rides. These aforementioned research fields were not included in this review because of the lack of a focus on sport and health. Furthermore, transportation papers and comparisons between regular bikes and e-bikes were also excluded, since they evaluate the e-bike’s potential benefits [35,36,37,38], while lacking the electronics and control integration. Some papers proposed communication methods using Internet of Things (IoT) protocols such as LoRa and XBee, as seen on Ambika bhuvaneswari and Muthumari [39] and Gharghan et al. [40]. The former presented an algorithm to reduce the energy consumption in a WSN (Wireless Sensor Network). Depending on the bike’s speed change, the algorithm augments or diminishes the data transmission. These articles have fundamental importance since they can reduce communication energy consumption, but they were considered “off topic“ for this review. Meanwhile, eco-mobility, air quality (measuring the fine particles in the air while commuting) and road optimization (choosing the best path depending on several factor such as distance, elevation, air quality, time, remaining battery, among others) papers were taken into consideration only if the proposed systems took into account the rider’s health in addition to the reduction in carbon footprint elements.

## 3. Results

The search provided 199 articles which were narrowed down to 114 after the elimination of duplicates. Eighty-six articles did not meet the inclusion criteria after titles, abstracts, and full-text readings. Thirty-four articles were not kept because they focused only on the electronic control units or battery management systems. Fifty-seven papers only examined asset the health and transportation aspects, and were, therefore, also excluded. The remaining 23 articles were considered.

### 3.1. Sensors

When discussing the sensors typically used on a bike, Global Positioning System (GPS), cadence in revolutions per minute [RPM], speed in kilometers per hour or mile per hour [Km/h, Mi/h], power in watts [W], and heart rate in beats per minute [BPM], are the first to be considered. BPM has been of the highest importance for health-focused studies because it is the primary indicator of exercise intensity [41,42,43] and is used to monitor diseases [44,45,46]. The other above-mentioned sensors are used as performance indicators, commonly used by professional athletes and amateurs to assist them in achieving their fitness or competitive goals. Additional sensors can be (and are) added to increase the amounts of environmental and physiological data, complement algorithms, or give more information to the user about its state and surroundings. Table 1 shows the sensors that are used the most often in the literature. As expected, heart rate is essential in this field, and almost every paper uses it to supervise the tester’s condition and adjust the algorithms in response. The selected papers used different technologies to measure the heart rate, such as Electrocardiogram (ECG), pulse oximeter (which also measures oxygen saturation), pulse sensor, chest belt, and self-made sensors.

Barrios et al. [70] compared a standard ECG with off-the-shelf wrist and armbands, concluding that wristbands were the most biased due to external interference. Etiwy et al. [71] showed that the chest belt monitor was in the strongest agreement with the ECG, significantly more so than the wrist band, whose accuracy varied depending on the type of exercise. Both articles have inferred that ECG is the preferred heart rate measuring method and should be used if possible. However, the downside of ECG is that it is not practical for outdoor use (patient comfort), which is where most of the subjects would perform tests (in real-life scenarios). The second and third most commonly used sensors measure power and cadence (or speed, since they can be deduced from each other). They are used in almost every motor control system since they retrieve the cyclist’s power or torque produced and their rate. This information correlated that the heart rate can be used to estimate the state of fatigue of the cyclist [52,53,57,65,68] as well as the road difficulty to assist the user in the best manner and reduce strain during their travels. GPS and accelerometer data are mainly used for road classification as slope recognition (flat, uphill, and downhill) and road segmentation [72,73], however they can be used in other applications (e.g., fall detection [74,75,76]).

### 3.2. Architectures

When it comes to the architecture of a smart electrical assisted bike, a computer is needed to recover and process the sensors’ data. Throughout the literature, systems may use one or several computers. Additional computers are used to complement the functionalities that one computer lacks or to increase the processing stages and augment computational power. Typically, the computers used on these systems are smartphones, servers, personal computers, and of course, microcontrollers. From Table 2, it can be inferred that microcontrollers are essential since they allow for the faster integration of sensors unable to communicate over the air, using protocols like BLE, WIFI, ZigBee, and Sigfox, to name a few. Microcontrollers also directly control and monitor the battery and the motors.

Nowadays, most people own a smartphone which can be exploited as a human machine interface (HMI). With these devices, users can parameter the algorithm and have visual updates of their physical activity in real-time [53,55,57,58,77]. It is possible to process the information and aid the controller in providing a better response, or transmit the collected data to a server or personal computer (PC) for further analysis [59,63,64,65,66,67]. Tablets could perform these functionalities but are limited by their size, which can be a nuisance on a bike. Moreover, some tablets don’t have an integrated GSM communication, or require the user to pay a subscription fee in order to use it.

Uploading data in real-time to a server is mainly used to retrieve, process, and analyze data, providing feedback to users [53,59,60,64,65,69] and allowing the health professionals to potentially monitor their patients remotely and adjust the algorithm (i.e., surveillance thresholds, training workload).

A significant majority of the studies included a smartphone as an HMI, which is essential for a sportive individual who wants to keep track of their data and post-activity recapitulation, or for a regular user who wants to travel from point A to point B. An interesting point is the potential of the HMI to personalize the bike, so that it responds accordingly to each rider, allowing them to introduce their profile (physiological characteristics) and desired training levels (e.g., time, distance, power). With an internet connection, the data can be reported on a corresponding website (server) so that healthcare professionals, coaches and individuals can visualize and follow activities, and adjust the workload.

### 3.3. Study Characteristics

Fifteen studies, reported in Table 3, have assessed their systems on 246 people. Higashi et al. [47], Ueno et al. [49] and De La Iglesia et al. [65] included in their studies a range of people between the ages of 20 to 30. Kiryu and Minagawa [52], De La Iglesia et al. [59] and Rey-Barth et al. [69] extended their studies to people in their sixties. On average, the studies included at least two testers, with the exception of [47,52,65,69] which included more than 9 subjects and [59] which included almost 190. The on-field tests were mostly comprised of flat and uphill segments, which had to be repeated multiple times while respecting specific test conditions such as keeping the same speed and/or cadence at all times. De La Iglesia et al. [59,65], Rey-Barth et al. [69] were the only ones that reported the duration of their studies, demonstrating the long-term impacts of these systems.

## 4. Discussion

As seen in Table 1, it is noticeable which sensors are used with electric bikes to transform them into health monitoring systems. Fundamentally, we can separate these sensors into three categories: physiological, neuromuscular, and environmental sensors.

Physiological sensors are more relevant for this purpose because they can investigate the physiological status of people during the ride and potentially estimate fatigue induced by exercise intensity and duration [78,79,80,81]. From a technical point of view, many sensors, that provide information related to the physiological values of the rider are available on the market [82]. Nevertheless, it is essential to consider the measurement process itself in order to maximize the user’s comfort level during the riding experience without disturbances that might lead to the loss of the electric assistance’s benefit. Secondly, it’s imperative to select values with the necessary background knowledge during the exercise to build an appropriate algorithm to model data or control loops to drive electric assistance.

The presented review pointed out that only heart rate (22/24 studies) and blood oxygen saturation (5/24 studies) were used for this purpose.

Heart rate was predominantly used for this application because of a large number of studies linking this parameter with health status and sport science experience legacy [78,79,80,83,84,85]. From a physiological aspect, heart rate is primarily used for its relationship with oxygen consumption (VO2), which is widely accepted as the gold standard to set exercise intensity. Based on Fick’s principle [86], cardiac output can be related to VO2, neglecting systolic ejection volume as demonstrated by Zhou et al. [87] with an elite athlete, trained, and untrained subject.

Based on that, several studies proposed a control model-based heart rate measurement using a preliminary test in a laboratory in order to get a better understanding of individual physiological states, using the heart rate and power outputs produced by the rider [57,68,88]; or used more basic methods such as the sliding mode controller [56].

The authors did not systematically describe the heart rate measurement process with sensor typology and location during the reviewed studies. Meyer et al. [88] used a chest strap in their study (Tickr; Wahoo Fitness, LLC, Atlanta, United States) to monitor heart rate during exercise, and we imagine that other studies used the same process because the chest belt is widely considered as the gold standard to measure heart rate accurately in numerous situations [71,89]. Recently, many new technical possibilities to monitor heart rate in non-invasive ways have appeared. They provide the most comfortable experience for people and have the potential for discrete integration in HMI [90]. Regarding e-bikes, few of those new possibilities are technically viable for implementation in a real ecological situation. In the last decade, many commercial heart rate monitors have implemented photoplethysmography technology as an alternative to traditional electrocardiography requiring chest belts, which may lead to discomfort or technical constraint for women, older, or obese people. This technique calculates heart rate by measuring micro blood flow oscillations in the skin. The working principle is based on the reflection of light passing through the skin [91,92]. Even if the technical aspect is promising, the actual accuracy of the commercial and experimental sensors is not sufficient for use with electrical assistance for healthy individuals or people with illnesses [71,89,93,94]. In his review for automotive sensors, Arakawa [90] describes the instrumentation potential of a steering wheel with bioelectrical measurement sensors that used the hands in order to detect a heartbeat. A similar system could be transposed on to the handlebar to eliminate third-party accessories, and measure the heart rate directly on a bike. Such sensors can be connected directly to an HMI, and an appropriate algorithm can be implemented to calculate the heart rate analogically, for example with other fitness equipment.

A consideration of heart rate measurement techniques is crucial for the future and possible implementation of heart rate sensors with more complex calculations as heart rate variability helps to estimate exercise intensity [95,96,97] for which accuracy is essential [98]. In addition, this potential implementation may lead to gaining an estimation of other different physiological variables such as breathing frequency [99], VO2 [100], and EPOC (Excess Post-Exercise Oxygen Consumption) [101]. Having this data will eventually result in more complex and dynamic models to control electrical assistance.

This review also points out that blood oxygen saturation may be used to control the stress caused by exercise [53,54] and also by environmental factors such as pollution [66]. Blood oxygen saturation is commonly used in the medical field to supervise patients in intensive care units and during the cardiac or pulmonary rehabilitation process [102]. This parameter provides more a profound insight into the individual’s physiological reaction during exercise. Nevertheless, blood oxygen saturation is obtained by an optical measurement that is more complicated to obtain than the heart rate’s. Traditionally, this optical measurement is performed at the finger but can also be measured at the ear, tympanic, or at several muscular locations [103]. This measurement is significantly affected by light and requires a light shield at the measurement location such as a finger hood or a large ear clip. For these reasons, it is not very easy to use these measurements in ecological tests, but depending on their evolution, these technologies could become an essential alternative for such measurements. Moreover, traditional finger measurement is affected by muscular contraction of the forearms and fingers when gripping the handlebar, controlling the bike trajectory, managing the braking system, and changing gear. As for the heart rate measurement, recent technical solutions have appeared and allow for the measurement of this variable with fewer constraints [93,104]. Several commercial wrist-worn products measure pulse oximetry using photoplethysmography while maintaining an affordable cost. Few researchers have investigated the accuracy of such devices for dynamic cycling, but recent studies from Schiefer et al. [105] and Hermand et al. [106] invite a careful consideration of these devices, especially when the pulse oximetry variable is of interest.

Today, it seems clear that heart rate remains the most practical solution for ecological utilization, and that more advanced devices would help get better algorithms for research purposes, but are not transferable in an actual application. As a gold standard, VO2 may be more used in the study of validation or proof of concept to get a clearer understanding of the physiological considerations and avoid heart rate shortcuts that may miss critical body reactions during specific field conditions. Gas exchange measurement with VO2 and VCO2 is widely used in the medical and sport science fields. This exam is accepted as the best indicator for determining the capacity of the heart, lungs, vessels, and muscles to effectively use oxygen during exercise ([107], pp. 1–3). This data could be collected from initial trials where subjects are given instructions on how to use the bike and its systems, how often it should be used, and on which path (if it is a controlled study). This would adequately customize the system to establish a rider’s intensity zone and evaluate if the subject should remain in that zone or change. Nagata et al. [51] and Sweeney et al. [66] measured VO2 and used it to supervise its evolution during a test, concluding that their system can prevent the user from crossing the second ventilatory threshold.

The actual state of the art for physiological assessments requires an oronasal face mask and gas exchange measurement system, which can be stationary and used in the laboratory, or be portable to evaluate people in ambulatory tests (actual field situations). It is of interest to maximize the utilization of VO2 measurement during such studies in order to upgrade the algorithms and increase the quality of metabolic predictions [108,109,110].

Regarding physiological measurements, we note that near infrared spectrometry is now widely used to grade physiological status [111] and may become more accurate and sensitive than heart rate in an ecological situations [112,113]. Today, these sensors are not ready to be integrated in the dynamic control of e-bikes, but may be good candidates in future works and add more relevant physiological information on leg muscles. This may give new insight on a potential control loop for electrical assistance based on local physiological markers on legs that could potentially exclude heart rate fluctuations related to side factors such as stress, temperature, caffeine, or others.

From general knowledge of the human body, it is well known that each person has their individual capabilities in both force and speed [114,115], leading to natural preferences during movements [116,117]. Moreover, the literature is well documented about the impact of strength limitations in cycling exercise within several populations, especially among those with diseases [118,119]. Based on this data, there may be a potential benefit for algorithms that adjust electrical assistance based on individual physiological and muscular (biomechanical) capacities in order to maintain people in their comfort zone.

In the same direction, the usage of electromyography (EMG) should be considered to assess the rider’s muscle fatigue during different terrain tests, i.e., assess the amount of time needed before fatigue is apparent on a flat, uphill, and mixed road, as seen in Kiryu and Minagawa [52]. Naturally, testing all the road combinations would take an enormous amount of time. However, this could help predict when and where the rider would need more assistance; helping to conserve the battery until it is needed. Furthermore, such tests can provide insights on potential muscular improvements for people who are in leg rehabilitation [120,121,122] ([123], p. 785).

A correlation must be determined using other physiological inputs (i.e., heart rate, cadence, and power) and muscular fatigue since it is not possible to have an EMG on the subject at all times when engaged in an outdoor session. However, this fatigue can be estimated using other inputs, as has been accomplished in works such as Corno et al. [57] and Meyer et al. [68]. Alternatively, even if this parameter can obtain relevant information to adjust the algorithm’s individualization accurately, it is essential to remember that surface muscular activation is not directly related to the force production ([124], p. 7). Moreover, the hardware setup and the experience of technicians need to be top tier to carefully place EMG sensors and manage the data. This typology of measurement can be possible in the laboratory. So far, few studies in the literature suggest that such devices can accurately and reliably measure the force or exercise induced fatigue [125,126].

One side effect of using an electric bike is that, while reducing the carbon footprint, it causes the cyclist to breathe polluted air, and this is noticeable in big cities ([127,128,129,130]) Sweeney et al. [66] and Venkatanarayanan et al. [67] proposed interesting ideas to control the respiration rate and measure air quality, respectively. Both are used to reduce the inhalation of traffic pollution, which has proven to have a greater negative impact on cyclists than people using other methods of transportation [131]. This is especially important to consider when the rider has respiratory pathologies such as asthma or even lung cancer. Caggiani et al. [132] addressed this issue by proposing an algorithm that can optimize the route depending on the user’s desires (i.e., travel time, safety, air pollution). Implementing this on an electrically assisted bike could help to protect the cyclist even further when commuting or doing exercise, and at the same time, collect data which can be used by local authorities and environmental groups to manage the air quality in cities and their surroundings, as proposed in works by Elen et al. [133], Liu et al. [134], Aguiari et al. [135], and Shen et al. [136].

The arrival of the 5G mobile networks will allow new technologies to surface, referring to the “5G triangle”, smart electrical bikes, in general, could fit in the mMTC (massive Machine Type Communication) use case. In a city with connected vehicles and city infrastructure, it is plausible to have bikes connected to that network as well, predicting the best route (considering traffic, bike lanes availability, air quality), and alerting other vehicles nearby, thus preventing more accidents. Meanwhile, environmental sensors can be recovered from the bike to be sent to a central hub, where this data could help improve weather prediction, air quality, and traffic reports. Health data recovered could be sent immediately, allowing health professionals to supervise their patients while doing rehabilitation, alerting them when it is necessary. Models used to predict health population could receive a boost by recollecting these on-field data, by analyzing the body’s response during an activity, providing new insights into how people with and without diseases perform, providing opportunities to prevent diseases by comparing these cases, and acting accordingly to improve the population’s health [137,138].

Good ergonomics are crucial to avoid injuries while using a bike. Bini et al. [139,140], Swart and Holliday [141] reviewed several works for saddle optimization intending to reduce injuries, concluding that a knee angle between 25° and 35° using the Holmes et al. [142] method is the most often used to reduce injuries. Nevertheless, further research is required to develop a thorough method that considers different saddle and handlebar positions, workload (watts, speed, cadence), and aerodynamics (sports performance). These new methods need to be tested in the field and on a static bike; position sensors can be placed on electric bikes and cyclists [143]. The information obtained can be compared in the field using different bike configurations so that pedaling techniques and riding styles can be identified and trained, as seen in Korff et al. [144] and Lucchetti et al. [145].

## 5. Conclusions

Electrically assisted bikes applied to the sport and health sector is a growing research field (exponentially growing in publications and number of citation) with direct and real applications that could change the perception of active mobility, and encourage many to start exercising progressively within their displacements, especially now that we are in a worldwide pandemic and people are reluctant to use public transportation.

Telemonitoring could be possible with every use; doctors and physicians could prescribe usage of the e-bike for the prevention of diseases like heart disease, strokes, diabetes, cancers, dementia, depression, and anxiety [13,146]. Using a web server or an app, health care professionals could connect to it and supervise patients at risk, following their activity and possibly having active communication with them, such as sending warning messages, increasing electric assistance, and changing the medical prescription (e.g., heart rate surveillance thresholds).

Test protocols for these kinds of devices are not yet properly established, which explains the discrepancies in the literature on the monitoring of physiological information and determining its purpose (i.e., pathology monitoring, physical therapy, active mobility, sports). Close collaboration with medical staff is imperative, so that the devices are validated both technological and clinically.

Nowadays, microcontrollers can integrate machine and deep learning libraries, al-lowing for more complex algorithms that can adapt depending on who is using the bike, and learn from the physiological and environmental data to better adjust electric assistance, battery management, and travel optimization. Further research is needed to conclude if such an intelligent system could have a relevant impact on the sportive and medical areas as well as providing an eco-friendly method of transportation.

## Figures and Tables

**Figure 1 sensors-22-00468-f001:**
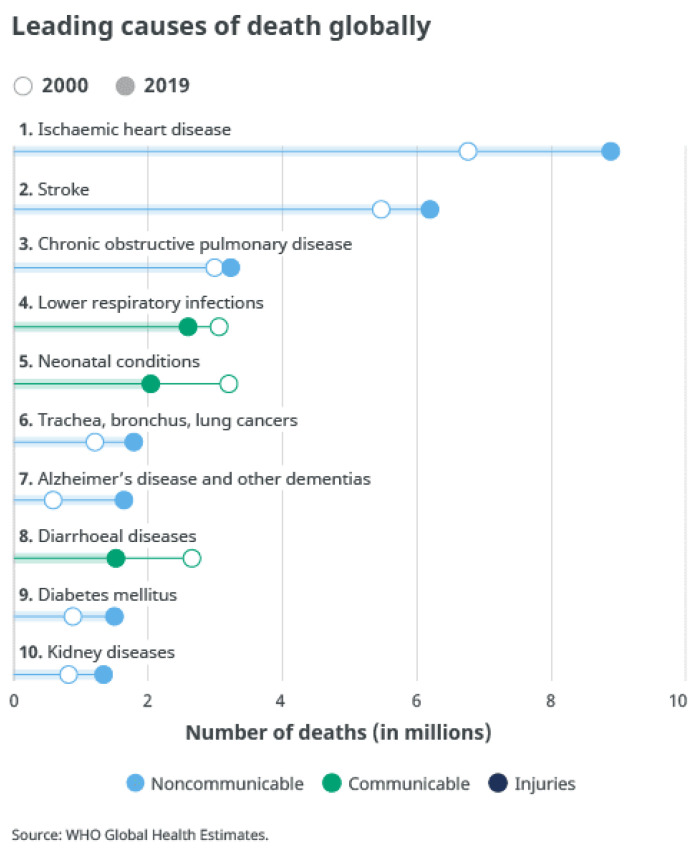
Top 10 causes of death [11].

**Figure 2 sensors-22-00468-f002:**
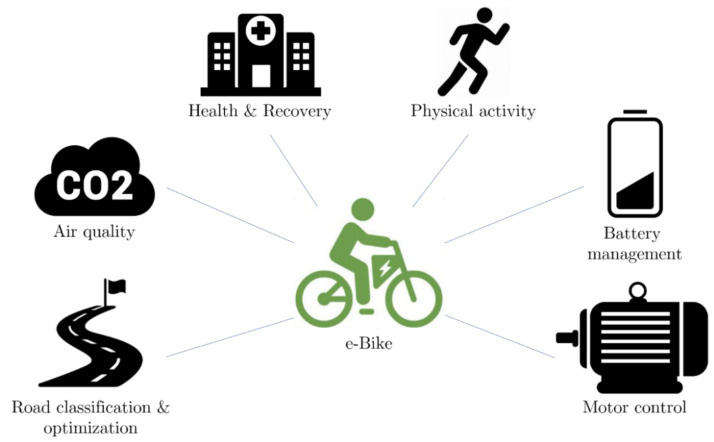
Electrically assisted bikes and associated research fields.

**Table 1 sensors-22-00468-t001:** Most common sensors used across the literature.

	Sensors	Oxygen Saturation	Heart Rate	Power/Force/Torque	Cadence	Speed	Gyroscope	Accelero-Meter	GPS	Weather
Articles										
A Wearable Capacitive Heart-Rate Monitor for Controlling Elec-trically Assisted Bicycle (2009) [47]		No	Yes	No	No	No	No	No	No	No
Mobile Health Monitoring Systems (2009) [48]		Yes	Yes	No	No	Yes	No	Yes	No	Tempera-ture
A wearable ECG-HR detector and its application to automatic assist-mode selection of an electrically as-sisted bicycle (2011) [49]		No	Yes	No	No	No	No	No	No	No
The design and implementation of the E-BIKE physiological monitoring pro-totype system for cyclists [50]		No	Yes	No	No	No	No	No	Yes	Tempera-ture
Electric Motor Assisted Bicycle as an Aerobic Ex-ercise Machine (2012) [51]		Yes (VO_2_)	Yes	No	Yes	No	No	No	No	No
Feasibility Study on a Perceived Fatigue Prediction Dependent Power Control for an Electrically Assisted Bicycle (2013) [52]*Inferred from each stroke using the electromyo-gram		No	Yes	*Yes	*Yes	*Yes	No	No	No	No
A Personalized and Context-Aware Mobile Assistance Sys-tem for Cardio-vascular Preven-tion and Rehabili-tation (2014) [53]		Yes	Yes	Yes	Yes	Yes	No	Yes	Yes	No
Optimization of Electric Bicycle for Youths with Disabilities (2014) [54]		Yes	Yes	Yes	Yes	Yes	No	No	No	No
Automatic Con-trol of Cycling Ef-fort Using Elec-tric Bicycles and Mobile Devices (2015) [55]		No	No	Yes	Yes	No	No	No	No	Yes
Human-in-the-Loop Bicycle Control via Ac-tive Heart Rate Regulation (2015) [56]		No	Yes	Yes	Yes	No	No	No	No	No
Design, Control, and Validation of aCharge-Sustain-ing Parallel Hy-brid Bicycle (2016) [57]		No	Yes	Yes	Yes	No	No	No	No	No
Health Analysis ofBicycle Rider andSecurity of Bicy-cle Using IoT (2017) [58]		No	Yes	No	No	Yes	No	Yes	No	No
Multi-Sensor InformationFusion for Opti-mizing Electric Bicycle Routes Using a Swarm Intelligence Algo-rithm (2017) [59]		No	Yes	No	Yes	Yes	No	No	Yes	No
Smart-Bike as One of the Ways to Ensure Sustainable Mobility in Smart Cities (2017) [60]		No	Yes	No	No	Yes	Yes	Yes	Yes	Yes
An Intelligent Control System for an Electrically Power Assisted Cycle (EPAC) (2018) [61]		No	Yes	No	Yes	Yes	No	Yes	No	No
Cyclist Monitoring System Using NI myRIO-1900 (2018) [62]		No	Yes	No	No	Yes	No	Yes	Yes	No
Development of Intelligent Smart Bicycle Control System (2018) [63]		No	Yes	No	No	Yes	Yes	Yes	Yes	Yes
HEALTHeBIKES–Smart E-Bike Prototype for Controlled Exercise in Tele rehabilitation Programs (2018) [64]		No	Yes	Yes	Yes	Yes	No	No	No	No
Increasing the Intensity over Time of an Electric-Assist Bike Basedon the User and Route: The Bike Becomes the Gym (2018) [65]		No	Yes	No	Yes	Yes	No	No	Yes	No
A context-aware e-bike system to reduce pollution inhalation while cycling (2019) [66]		Yes	Yes	Yes	No	Yes	No	No	No	Yes (air quality)
Design of sensor system for air pollution and human vital monitoring for connected cyclists (2019) [67]		No	Yes	No	No	No	No	Yes	No	No
Regulating the Heart Rate of Human–Electric Hybrid Vehicle Riders Under Energy ConsumptionConstraints Using an Optimal Control Approach (2019) [68]		No	Yes	Yes	Yes	No	No	No	No	No
Use of a smart electrically assisted bicycle (VELIS) in the health field Proof of concept (2020) [69]		No	Yes	Yes	Yes	Yes	No	No	Yes	No

* They used an electromyogram to evaluate the fatigue on the leg muscles, and from that measure, they also estimate the values of torque, cadence and speed.

**Table 2 sensors-22-00468-t002:** Architectures used across the literature.

	Architectures	PC	Server	Micro-Controller	Smartphone
Articles					
A Wearable Capacitive Heart-Rate Monitor for Controlling Electrically Assisted Bicycle(2009) [47]		No	No	Yes	No
Mobile Health Monitoring Systems (2009) [48]		Yes	Yes	Yes	No
A wearable ECG-HR detector and its application to automatic assist-mode selection of an electrically assisted bicycle (2011) [49]		Yes	No	No	No
The design and implementation of the E-BIKE physiological monitoring prototype system for cyclists (2011) [50]		No	No	Yes	No
Electric Motor Assisted Bicycle as an Aerobic Exercise Machine (2012) [51]		No	No	Yes	No
A Personalized and Context-Aware Mobile Assistance System for Cardiovascular Prevention and Rehabilitation (2014) [53]		Yes	Yes	Yes	Yes
Optimization of Electric Bicycle for Youths with Disabilities (2014) [54]		Yes	No	Yes	No
Automatic Control of Cycling Effort Using Electric Bicycles and Mobile Devices (2015) [55]		No	No	Yes	Yes
Human-in-the-Loop Bicycle Control via Active Heart Rate Regulation (2015) [56]		No	No	Yes	No
Design, Control, and Validation of aCharge-Sustaining Parallel Hybrid Bicycle (2016) [57]		No	No	Yes	Yes
Health Analysis ofBicycle Rider andSecurity of Bicycle Using IoT (2017) [58]		No	No	Yes	Yes
Multi-Sensor InformationFusion for Optimizing Electric Bicycle Routes Using a Swarm Intelligence Algorithm (2017) [59]		No	Yes	Yes	Yes
Smart-Bike as One of the Ways to Ensure Sustainable Mobility in Smart Cities (2017) [60]		No	Yes	Yes	No
An Intelligent Control System for an Electrically Power Assisted Cycle (EPAC) (2018) [61]		No	No	Yes	No
Cyclist Monitoring System Using NI myRIO-1900 (2018) [62]		Yes	No	Yes	No
Development of Intelligent Smart Bicycle Control System (2018) [63]		No	No	Yes	Yes
HEALTHeBIKES–Smart E-Bike Prototype for Controlled Exercise in Tele rehabilitation Programs (2018) [64]		No	Yes	Yes	Yes
Increasing the Intensity over Time of an Electric-Assist Bike Basedon the User and Route: The Bike Becomes the Gym (2018) [65]		No	Yes	Yes	Yes
A context-aware e-bike system to reduce pollution inhalation while cycling (2019) [66]		No	No	Yes	Yes
Design of sensor system for air pollution and human vital monitoring for connected cyclists(2019) [67]		No	No	Yes	Yes
Regulating the Heart Rate of Human–Electric Hybrid Vehicle Riders Under Energy ConsumptionConstraints Using an Optimal Control Approach (2019) [68]		No	No	Yes	No

**Table 3 sensors-22-00468-t003:** Study characteristics across the literature.

	Characteristics	Subjects (Sex)	Mean Age	Mean Weigh	Mean Height	Activity	Condition	Duration
Articles								
A Wearable Capacitive Heart-Rate Monitor for Controlling Electrically Assisted Bicycle (2009) [47]		4 (N/S)	21.8	60.3	170	1.14 Km on flat road and 365 m of 4% slope	2 min rest prior to the activity. Constant speed of 10 Km/h	
A wearable ECG-HR detector and its application to automatic assist mode selection of an electrically assisted bicycle (2011) [49]		1 (M)	22			4 loops–224 m flat and 106 m slope–3 sets: one without and with assistance, and the proposed control	Constant speed of 12 Km/h	
Electric Motor Assisted Bicycle as an Aerobic Exercise Machine (2012) [51]		5 (N/S)				4.6 Km mixed road–3 sets: assistance control at 65 RPM, 70 RPM and regular assistance		
Feasibility Study on aPerceived Fatigue Prediction Dependent Power Control for an Electrically Assisted Bicycle (2013) [52]		17 (12M, 5F)	8M (23.8 ± 2.3) 4M (61.3 ± 8.1) 5F (44.2 ± 6.3)			2.1 Km with 600 m of uphill–20 min rest intervals	Constant cadence of 60 RPM	
Optimization of Electric Bicycle for Youths with Disabilities (2014) [54]		1 (N/S)				2 sets: with assistance and without (on a home trainer)		
Automatic Control of Cycling Effort Using Electric Bicycles and Mobile Devices (2015) [55]		1 (N/S)				Home trainer resistance minimum value for 30 s. From 30 to 50 s resistance increased to maximum for 70 s. After resistance was gradually decreased–2 sets: with assistance and without	Constant cadence	
Human-in-the-Loop Bicycle Control via Active Heart Rate Regulation (2015) [56]		2(N/S)				2 sets: Traditional mode and HR control mode		
Design, Control, and Validation of a Charge-Sustaining Parallel Hybrid Bicycle (2016) [57]								
First study		1 (N/S)				1.1 Km with an average speed of 20 Km/h–2 sets: without assistance and with the algorithm		
Second study		2 (N/S)				20 min of route with ± 5% slopes–2 sets: without assistance and with the algorithm		
Multi-Sensor Information Fusion for Optimizing Electric Bicycle Routes Using a Swarm Intelligence Algorithm (2017) [59]		187 (N/S)	Between 17 and 58			3.82 Km route (home to work)		11 Months
Cyclist Monitoring System Using NI myRIO 1900 (2018) [62]		1 (N/S)				Standard procedure		
HEALTH eBIKES–Smart E-Bike Prototype for Controlled Exercise in Tele rehabilitation Programs (2018) [64]		2 (N/S)				7 Rides on different terrains		
Increasing the Intensity over Time of an Electric Assist Bike Based on the User and Route: The Bike Becomes the Gym (2018) [65]		9 (6M, 3F)	30.9 ± 4.7					4 Months
A context-aware e-bike system to reduce pollution inhalation while cycling (2019) [66]		1 (N/S)				5 Km mostly flat route–2 laps	Constant speed of 20 Km/h	
Regulating the Heart Rate of Human–Electric Hybrid Vehicle Riders Under Energy Consumption Constraints Using an Optimal Control Approach (2019) [68]						Validation through simulation		
Use of a smart electrically assisted bicycle (VELIS) in the health field Proof of concept (2020) [69]		12 (7M, 5F)	Between 29 and 66			14 Km with 350 positive elevation (6 Km flat, 2.5 Km uphill and 5.5 Km downhill)–4 sessions	Constant cadence of 55, 65 and 75 RPM depending on the day	One week

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
