# Peer review of "Smart Electrically Assisted Bicycles as Health Monitoring Systems: A Review"

_sensors, 2022, doi:10.3390/s22020468_

Round 1

Reviewer 1 Report

The following are my conclusions:
- the paper describes an interesting topic;
- the contributions of the paper are based on realistic and referenced assumptions;
- the problem in the manuscript is well defined, and the objectives are clear;
- the paper adequately put the progress it reports in the context of previous works, representative referencing and introductory discussion
- the conclusions and potential impacts of the paper are made clear. 

Increase the bibliography:

bibliographic suggestions
A Novel Acceleration Signal Processing Procedure for Cycling Safety Assessment
E Murgano, R Caponetto, G Pappalardo, SD Cafiso, A Severino
Sensors 21 (12), 4183

Author Response

Dear reviewer,  

First, we want to thank you for taking the time to asses our paper.  

We have taken into consideration your suggestion's and we have indeed added more bibliography  

Murgano et Al. details a novel signal processing method to correct acceleration and speed data collected by embedded sensors. At the same time, the proposed method estimates hard braking for cyclist safety assessment. 

Because of the importance of cyclist safety in medical applications for e-bikes, it seems important to authors to consider the Murgano et Al. procedure in current review.

Looking forward to hear from you.

Best regards,

Eli Gabriel AVINA

Reviewer 2 Report

The authors reviewed the electrically assisted bicycles (e-bikes) which is used to recover the rider’s physical and physiological information, monitor their health 15 state, and adjust the "medical" assistance accordingly. Meanwhile the need is deeply discussed that Artificial Intelligence (AI) systems that recovers and processes a large amount of data will be equipped with the e-bikes. The systematic review are completed from The Preferred Reporting Items for Systematic Reviews and Meta-Analyses Guidelines by relevantly being searched and selected. However, I suggest this manuscript with a major revision for the submission according to the following concerns:

  • In Fig. 2, electrical assisted bikes and associated research fields are introduced. Then most common sensors, architectures and study characteristics used across the literature are introduced and compared. However, the differences of their relevant technical routes about the implement of e-bikes are not indicated in details, and relevant additional technology roadmaps may be in need for the readers to read more easily.
  • There are a few grammatical errors in this manuscript.

Author Response

Dear reviewer,  

First, we want to thank you for taking the time to asses our paper.  

We have taken into consideration your suggestion's and we have explained more the different technological routes (in 2.2 Inclusion criterial), especially the ones that are not very present on the paper because they considered off topic.

Looking forward to hear from you.

Best regards

Eli Gabriel AVINA

Reviewer 3 Report

The paper aims to be a survey of ebikes as tools for health monitoring. However, in my opinion its scope is unclear. The authors started to look for references based on a number of keywords and concepts related to ebikes. As they point out in Fig. 2 most of these are not related to health. Even though the ruled out most of the papers it still seems the paper that have left are a very heterogenous group. The authors characterize sensors used and data transfer/processing solutions as well as other functions of systems/approaches described in the papers. However, from the discussion it is still very difficult to learn what valuable contributions are present in the papers they mention. Moreover, it seems number of results related to health they describe seem to be disconnected from ebikes as such.

Finally, the paper is of bad linguistic and editorial quality.

Author Response

Dear reviewer,

First, we want to thank you for taking the time to asses our paper.

The purpose of this article is twofold:

  • Propose a literature review on e-bikes with medical or health monitoring applications,
  • On the systems listed in the literature, the authors carried out a comparative analysis of the technological choices made on the different subsystems (sensors, architecture, HMI, etc.).

On the first point:

The authors regret that, after consulting databases such as PubMed and Web of Science, few articles attest to the use of an electrically assisted bike coupled with connected objects in clinical studies.

Nevertheless, this confirms the hypothesis in the interest to develop a medical device such as an electrically assisted bike equipped with an AI to allow a healthcare service with close collaboration with medical teams.

On the second point:

The literature extracted so far shows very diverse applications for electrically assisted bikes, and not only medical. Furthermore, the validation methods used are not standardised on the published articles, which can lead to bias if they are used to validate those devices for medical purposes (analysis of the benefits and risks for the patients' health).

For this reason, the authors sought to compare the literature from a technological point of view, since all existing systems require sensors and electronics architectures to acquire and process the data (Table 1 and Table 2).

Finally, in Table 3, the authors cross-referenced the validation methods used on human subjects.

The perspectives of this work (of which this article is only a first step) will be in the conception of a medical electrically assisted bike with AI, but also to express the need to have a standardised method of clinical validation for such an intelligent e-bike.

The authors hope to have enlightened the main purpose of this article, and their overall scientific approach.

Looking forward to hear from you.

Best regards,

Reviewer 4 Report

This paper provides a review of the e-bikes for recovering the rider’s physical and physiological information, monitor their health state, and adjust the "medical" assistance accordingly.

E-bikes make cycling less of a physical effort, opening it up to many who might have hesitated before. Meanwhile, technology innovations including predictive analytics, wireless connectivity, digital urban planning tools  will launch with the aim of making cycling safer, faster, more convenient, and easier to track and measure.

The authors should include the role of 5G in the e-bikes revolution. 5G networks can support faster data speeds, lower latency and greater connectivity to keep a city green and people safe as they travel.

Author Response

Dear reviewer,  

First, we want to thank you for taking the time to asses our paper.  

We have taken into consideration your suggestion's and we have added a paragraph (in 4. Discussion) where we discuss how the 5G could change and improve the existing systems.

Looking forward to hear from you.
Best regards,

Round 2

Reviewer 1 Report

Good work.

okay !

for me is publication!!

Reviewer 2 Report

  The authors  have taken into consideration my suggestions and then solved the questions that I have proposed by explaining more the different technological routes in the inclusion criterial part of Subsection 2.2, and especially the ones that have been considered off topic and not very present in the paper. As a result, I suggest this manuscript accepted for the submission.